**Subject Area:**
biophysics/structural biology

meiosis, homologous recombination, synaptonemal complex, SYCP3, cryoEM, AFM

**Authors for correspondence:**
Ioanna Mela
e-mail: im337@cam.ac.uk
Luca Pellegrini
e-mail: lp212@cam.ac.uk

[†]These authors contributed equally to this work.

# Molecular architecture of the SYCP3 fibre and its interaction with DNA

Daniel Bollschweiler[1,†], Laura Radu[1,†], Luay Joudeh[1], Jürgen M. Plitzko[2], Robert M. Henderson[3], Ioanna Mela[3] and Luca Pellegrini[1]

[1]Department of Biochemistry, University of Cambridge, Cambridge CB2 1GA, UK
[2]Max Planck Institute of Biochemistry, 82152 Martinsried, Germany
[3]Department of Pharmacology, University of Cambridge, Cambridge CB2 1PD, UK

(iD) IM, 0000-0002-2914-9971; LP, 0000-0002-9300-497X

The synaptonemal complex (SC) keeps homologous chromosomes in close alignment during meiotic recombination. A hallmark of the SC is the presence of its constituent protein SYCP3 on the chromosome axis. During SC assembly, SYCP3 is deposited on both axes of the homologue pair, forming axial elements that fuse into the lateral element (LE) in the tripartite structure of the mature SC. We have used cryo-electron tomography and atomic force microscopy to study the mechanism of assembly and DNA binding of the SYCP3 fibre. We find that the three-dimensional architecture of the fibre is built on a highly irregular arrangement of SYCP3 molecules displaying very limited local geometry. Interaction between SYCP3 molecules is driven by the intrinsically disordered tails of the protein, with no contact between the helical cores, resulting in a flexible fibre assembly. We demonstrate that the SYCP3 fibre can engage in extensive interactions with DNA, indicative of an efficient mechanism for incorporation of DNA within the fibre. Our findings suggest that SYCP3 deposition on the chromosome axis might take place by polymerization into a fibre that is fastened to the chromosome surface via DNA binding.

## 1. Introduction

At the heart of meiosis lies the exchange of genetic material between homologous parental chromosomes, which generates sexual diversity in the offspring. Correct and efficient recombination (crossover) between homologous chromosomes requires that they become physically aligned along their length. A supramolecular proteinaceous structure known as the synaptonemal complex (SC) is a critical participant in the process of crossover at meiotic prophase, by acting as a molecular zipper and linking the homologous chromosomes during recombination [1,2]. Despite considerable variation in the nature and sequence of its protein components, the SC structure shows a remarkable evolutionary conservation, consisting of lateral elements (LEs) that form on each paired chromosome and are linked by transverse filaments to a central element (CE) [3]. Relatively little is known about the detailed structure of the SC, and how its dynamic architecture influences meiotic processes. Recent studies using super-resolution fluorescence have begun to provide important insights into the overall 3D structure of the SC and location of known factors [4–6].

SYCP3 is a major protein component of the LE [7]. It is recruited to meiotic chromosomes in leptotene, contributing to the formation of axial elements that spread over the entire length of the chromosome axis at pachytene, resulting eventually in the formation of the LE in the tripartite SC structure [8]. The molecular mechanism of LE assembly is unknown, but it requires the involvement, together with SYCP3, of other meiotic axial proteins. Indeed, an evolutionarily conserved set of chromosome axis proteins comprising members of the Hormad family and coiled-coil proteins (for example, Hop1 and Red1 in yeast, ASY1 and ASY3, ASY4 in plants, Hormad 1, 2 and SYCP2, SYCP3 in humans) is likely to be responsible

for the assembly and structure of the LE [9–11]. The three-dimensional architecture of the meiotic chromosome that is competent for recombination results from the functional and physical interactions of SC components with meiotic cohesin complexes [12–17].

Correct SYCP3 function is essential for SC formation, chromosome synapsis and fertility [8,18]. *Sycp3* gene knockout causes infertility in male mice and reduced fertility in females, due to aneuploidy in oocytes and ensuing embryonic lethality [8,19], and a mutation that renders SYCP3 defective causes human male infertility [20]. Overexpression of SYCP3 has been reported in some types of cancer [21,22]. In SYCP3-null male mice, the LE and SC do not form and homologous chromosomes fail to achieve full synapsis [8,17,23]. A twofold increase in chromosome fibre length in SYCP3-deficient mice oocytes suggests a defect in chromosome organization [19]. Presumably, the LE must assemble on the existing structure of the chromosome axis, which is determined to a large extent by meiotic cohesins [14,15,24,25].

SYCP3 folds into a highly elongated helical tetramer, where each chain forms antiparallel coiled-coil interactions and with two N- and C-tails protruding at each end of the helical core [26]. A well-characterized property of SYCP3 is its ability to form filamentous fibres displaying transversal striations when overexpressed in mammalian cells [20,27–29]; this behaviour is mirrored by the ability of the recombinant protein to polymerize into similar, striated filamentous structures [26]. Although the structural determinants driving SYCP3 polymerization are presently unknown, self-assembly is critically dependent on sequence motifs in the N- and C-termini of the SYCP3 tetrameric structure [26,28].

In addition to forming large filamentous structures, SYCP3 can interact with double-stranded DNA, via DNA-binding motifs located in its N-terminal tails [26]. The presence of DNA-binding domains at either end of the elongated rod-like shape of the SYCP3 tetramer indicates that it can simultaneously interact with distinct segments of DNA [30]. Single-molecule studies of the interaction of SYCP3 with DNA showed that DNA-bound SYCP3 molecules can form clusters that drive a limited degree of DNA compaction, in agreement with SYCP3's structural role in determining LE architecture [30].

In this paper, we set out to elucidate how SYCP3 self-associates into filamentous fibres and how the SYCP3 fibres interact with DNA. Using a combination of cryo-electron tomography of the SYCP3 fibres and atomic force microscopy of SYCP3-DNA complexes, we have obtained important new insights into SYCP3 function. We show that the regular higher-order structure of the SYCP3 fibre arises from a remarkably heterogeneous mode of association of individual SYCP3 particles, conferring plasticity to the fibre. Furthermore, we provide experimental evidence that polymeric SYCP3 fibres can engage in extensive interactions with DNA. Our results suggest that SYCP3 can coat the chromosome axis in a continuous structure containing both DNA-bound and DNA-free SYCP3 layers. We discuss the implications of this structural model for the function of SYCP3 and LE assembly in meiosis.

# 2. Material and methods

## 2.1. Protein expression

Amino acid sequences corresponding to 1–236 (full length) and 1–230 of human SYCP3 were cloned into the pHAT4 vector [31] for expression in bacteria with an N-terminal TEV-protease cleavable His6-tag. Recombinant proteins were expressed in Rosetta 2 (DE3) *E. coli* (Novagen). Transformed cells were plated out on LB agar supplemented with 34 µg ml$^{-1}$ chloramphenicol and 100 µg ml$^{-1}$ ampicillin. Bacterial colonies were transferred to 1 l 2xYT (100 µg ml$^{-1}$ ampicillin, 34 µg ml$^{-1}$ chloramphenicol) at 37°C and grown until 0.6 OD600 in a baffled 2 l flask. Bacteria were then induced (0.5 mM IPTG, 25°C) and grown overnight, harvested by centrifugation (4000$g$, 20°C, 20 min), resuspended in 20 mM Tris-HCl pH 8.0, 400 mM KCl buffer (25 ml buffer per pellet of 1 l culture) containing EDTA-free protease inhibitors (Sigma) and stored at −80°C.

## 2.2. Protein purification

The bacteria were lysed by sonication, the lysate was clarified by centrifugation (30 000$g$, 4°C, 30 min) and passed through a 45 µm filter. SYCP3 proteins were first purified using Ni-NTA agarose resin (Qiagen). A Ni-NTA column (2 ml) was equilibrated with five column volumes of wash buffer (20 mM Tris pH 8.0, 400 mM KCl). The clarified lysate was passed over the column under gravity flow. The column was washed with 20 column volumes of wash buffer to remove unbound bacterial proteins, followed by 4 column volumes of wash buffer with 20 mM imidazole to remove weakly binding bacterial contaminant proteins. Bound SYCP3 was recovered in two elution steps, using wash buffer with 100 and 200 mM imidazole. The 200 mM imidazole elution typically contained the cleanest recombinant protein, and was purified further as described below.

### 2.2.1. Full-length SYCP3

To cleave the His$_6$-tag, TEV protease was added at mass ratio of 1 : 50 and incubated at 4°C overnight. To help remove nucleic acid contamination present after the Ni-NTA step, benzonase was also added during the TEV treatment. Full-length SYCP3 was then concentrated by centrifugation device and further purified by gel filtration over a Superdex 200 16/60 column equilibrated in 20 mM Tris-HCl pH 8.0, 400 mM KCl. Peak fractions were analysed by SDS-PAGE, pooled, spin concentrated (4500$g$, 10°C) using 4 ml Amicon Ultra 10 000 MWCO filter units (Millipore), flash frozen and stored at −80°C.

### 2.2.2. SYCP3 1–230

To cleave the His$_6$-tag, TEV protease was added at mass ratio of 1 : 50 and incubated overnight on ice. SYCP3 1–230 was further purified by cation-exchange over a 5-ml HiTrap SP column (GE Healthcare) and Heparin Sepharose chromatography (GE Healthcare), using a linear salt gradient from 150 mM to 1 M KCl gradient in 20 mM Tris-HCl pH8.0, 150 mM KCl. Peak fractions were pooled, spin concentrated (4500$g$, 10°C) using 4 ml Amicon Ultra 10 000 MWCO filter units (Millipore), flash frozen and stored at −80°C.

## 2.3. Salt-dependent aggregation of SYCP3

The aggregation index (AI) of the SYCP3 protein was analysed adapting a published protocol [32], using the following formula:

$$AI = 100 \times \frac{A340}{A280 - A340},$$

where A340 and A280 are the absorbance of the SYCP3 sample at 340 nm and 280 nm, respectively. The AI provides an indication of the aggregation state of a protein sample in solution, independent of its concentration. The AI was used to study the salt dependency of SYCP3 aggregation, measuring five SYCP3 samples at 25 µM in 20 mM Tris-HCl buffer pH 8.0 and KCl concentrations ranging from 100 to 500 mM.

## 2.4. Sample vitrification for cryo-electron tomography

100 µl of BSA-coated 10 nm gold bead suspension (Aurion, Wageningen, The Netherlands) were centrifuged at 14 000$g$ for 10 min, the clear supernatant discarded, and the gold beads resuspended in 50 µl ice-cold 20 mM Tris, pH 8.0 (DB buffer). Quantifoil grids (Cu 200 mesh, 2/1 holey carbon, Quantifoil Micro Tools GmbH, Jena, Germany) were plasma-cleaned for 30 s. Vitrification was carried out in a Vitrobot Mk IV (FEI/Thermo Fisher, Eindhoven, The Netherlands), with the sample chamber set to 4°C and 90% humidity, a blotting strength parameter set to '10' and a blotting time of 5 s. 3 µl of previously prepared gold particle resuspension were applied to the grid. The droplet was allowed to dry for about 2 min, until only a thin liquid film covered the grid. Immediately afterwards, 4 µl of ice-cooled SYCP3 protein solution (14 mg ml$^{-1}$), freshly mixed 1 : 1 with DB buffer, to reduce salt concentration and promote fibre formation, were pipetted onto the grid. After a short incubation time of 10 s, the grid was transferred to the Vitrobot, blotted and plunged into a liquid nitrogen-cooled bath of 37% ethane, 63% propane.

## 2.5. Cryo-electron tomography of SYCP3 fibres

Ten tilt series of fibres were recorded with a Tecnai G2 Polara (Thermo Fisher), equipped with a 300 kV FEG, a Gatan post-column energy filter and a K2 Summit direct electron detector (Gatan). Images were recorded at a working magnification factor of 34 000×, a pixel size of 3.45 Å px$^{-1}$ and a dose rate of 0.8 e Å$^{-2}$ s$^{-1}$ with a total dose per tilt series of 70–100 e Å$^{-2}$. The target defocus was 2.5 µm. For each tilt series, 61 micrographs were recorded at 2° increments with a tilt order of −30° to +60° followed by −32° to −60°.

## 2.6. Tomographic reconstruction and subtomogram analysis

Here we provide a complete description of the subtomogram analysis of the SYCP3 fibres under native conditions. Further information on the method can be found in our recently published protocol for analysis of filamentous protein assemblies such as the SYCP3 fibres [33]. Drift correction for the tilt series micrographs was performed with MotionCor2 [34]. The tomograms were backprojected in IMOD [35], using the embedded 10 nm gold markers as fiducials for the alignment. Due to the characteristically low signal-to-noise of the tomographic reconstructions, it was not practical to identify and select individual SYCP3 tetramers as subvolumes. Instead, subvolume coordinates were picked along the striation density of the fibres. To facilitate the detection of the striations within the fibre, the tomographic volumes were low-pass filtered with a cut-off at 0.44 nm$^{-1}$. Two coordinate models were created using IMOD [35], by manually tracing the ends of individual striations with a single contour and

then interpolating equidistant points within each contour via the program addModPts of the PEET package [36,37]. The first model featured a smaller set of 5756 coordinates, widely spaced at 44 nm (128 voxels), which was used to generate an initial reference structure *de novo* (see below). The second model comprised a total of 46051 subvolume coordinates, at a narrow spacing of 16.5 nm (48 voxels) in the *XY* plane and 22 nm (64 voxels) along *Z*. This set of coordinates was used later for masked structural refinement.

Subtomogram averaging was performed in RELION v. 2.1, adapting a modified protocol by Bharat & Scheres [38]. The IMOD coordinate models were converted to ASCII format using model2point [35]. CTF estimation and correction was performed with ctffind4 (4.0.15) [39], via the relion_prepare_subtomograms.py script [38]. The first set of subvolume coordinates was used to extract 5767 non-overlapping subvolumes, rescaled from 128$^3$ voxels unbinned to a size of 64$^3$ voxels (equivalent to 44 × 44 × 44 nm$^3$ at a pixel size of 6.9 Å px$^{-1}$). An identical set was extracted and resized with a binning factor of 2. Due to the manual picking of coordinates, each subvolume was only roughly centred on a striation density. Since the long axis of the fibres was oriented parallel to the grid surface, a translational alignment in the *XY* plane was sufficient to precisely align all subvolumes within the fibre pattern. For this, *Z*-projections of all rescaled subvolumes were used as the input for a reference-free 2D classification of three classes. This step served as an *XY*-plane alignment measure, as well as a screening step to remove unsuitable particles, such as subvolumes picked too close to the edge of a fibre, from the following processing steps. The majority of particles (51%) contributed to a class that showed a clear striation density layer at its centre, with separate elongated densities of around 2 × 20 nm in size connecting orthogonally to adjacent striations, in good agreement with the dimensions of the coiled-coil core region of an SYCP3 tetramer [26]. Using the relion_2Dto3D_star.py script [38], the file paths of 2D projections contributing to this class average were translated back to their respective 3D subvolumes in a new particles_subtomo_2Dto3D.star table. In addition to the file paths, the translational shift columns (_rlnOriginX and _rlnOriginY) were added for each subvolume. Using this list of *XY*-centred subvolumes, a reference-free 3D angular search was performed. The resulting single-class average constituted a low resolution *de novo* structure and was rotated with IMOD's rotatevol function (http://bio3d.colorado.edu/imod/doc/man/rotatevol.html) so that the striation plane was orthogonal to the *Y*-axis of the subvolume. The rotated structure was subsequently refined in 3D classification and 3D auto-refinement steps against the entire first set of subvolumes. The refined structural average featured a clearly aligned centred striation density, with orthogonally connecting elongated SYCP3 tetramers, roughly arranged in a rectangular mesh along the striation plane.

Using the second set of coordinates, 46051 overlapping subvolumes of 44 × 44 × 44 nm$^3$ (128$^3$ voxels at the unbinned pixel size of 3.45 Å px$^{-1}$) were extracted and rescaled by a factor of 0.5, corresponding to a subvolume size of 64$^3$ voxels at twice the pixel size (6.9 Å px$^{-1}$). This larger set of subvolumes was used for a masked 3D classification, with the refined *de novo* structure created in the previous step serving as the input reference. The reference mask was created in Matlab/Tom Toolbox [40], and consisted of a narrow cylinder of 24 voxel diameter (equivalent to 16.5 nm at a binning factor of 2) with its long axis oriented parallel to the *Y*-axis and a

smooth Gaussian falloff within a rescaled $64^3$ voxel volume. The masked 3D classification resulted in 4 class averages, with nearly equal class distributions. Subvolumes with a large translational shift (with individual translational shifts in $X$, $Y$ and $Z \geq d/(2\sqrt{2})$, $d$ being the distance of subvolume coordinates in voxels) were removed to exclude auto-correlation effects of potentially overlapping subvolumes. Each trimmed class dataset was then 3D auto-refined.

These final refined class averages showed a high degree of heterogeneity in the organization of the SYCP3 tetramer within the fibre, with resolutions of around 30–27 Å. Additional hierarchical 3D classification of these classes, focusing on a single striation-striation repeat by using a reference mask of only a half cylinder and splitting each refined subvolume subset into 8 classes, resulted in a clearer separation of structural motifs but did not improve resolution. Consistent with this, the class distribution was close to 12.5% (±2%) for the refined subsets 1, 2 and 3, confirming the presence of high structural heterogeneity; subset 4 showed a slight deviation from this pattern as it featured two heavily warped class averages with 16.6% and 15.9% subvolume distribution, probably due to the missing wedge-induced elongation of subvolumes, which could not be compensated for with sufficient precision.

## 2.7. Atomic force microscopy

For analysis of the DNA-free fibres, purified SYCP3 was diluted to micromolar concentrations (1–3 μM) into a low-salt buffer (20 mM Tris-HCl pH 8.0, 150 mM NaCl) to induce fibre formation, deposited on freshly cleaved mica and incubated at room temperature for 10 min. The samples were washed five times with 1 ml of BPC grade water (Sigma) and dried under a nitrogen stream. For fibre formation in the presence of DNA, 0.5 nM plasmid DNA was mixed with a 5-fold molar excess of SYCP3 in the low-salt buffer and incubated for 5 min at room temperature. Before deposition on mica, 5 mM $MgCl_2$ were added to the buffer, to facilitate binding of the protein–DNA complexes on the negatively charged mica surface. The experiments were also performed on poly-l-lysine coated mica in the absence of $MgCl_2$, with identical results. The majority of experiments were performed with pUC19 DNA, but other circular double-stranded DNA such as PhiX174 RFII were tested with similar results. The samples were washed and dried as described above. Imaging was performed using a Bruker Dimension FastScan AFM, in air, at ambient temperature. The probes used were FastScan-A (Bruker), with resonant frequency of 1400 kHz and the drive frequencies were approximately 5% below the maximal resonance peak. AFM images were acquired at a rate of four frames per minute and plane-fitted to remove tilt. Each scan line was fitted to a first-order equation, using Nanoscope Analysis v. 1.9 software (Bruker). To measure the height of the fibres, cross sections were taken across ten different areas of each fibre using Nanoscope Analysis. The length of fibres reported, represents the longest available continuous path along the fibre.

# 3. Results

## 3.1. *In vitro* reconstitution of the SYCP3 fibres

We had previously shown that recombinant human SYCP3 forms filamentous fibres that can be visualized by negative-stain electron microscopy (EM) on continuous carbon grids [26]. The fibres show a periodic pattern of transversal, alternating light and dark bands, appearing at regular intervals of 22 nm. To determine whether SYCP3 fibre formation and its periodic appearance is an intrinsic property of SYCP3 in solution, we examined recombinant SYCP3 by cryo-electron microscopy. First, we established buffer conditions for controlled SYCP3 polymerization, by determining that salt concentrations less than 300 mM KCl drive rapid SYCP3 self-association into a fibre (electronic supplementary material, figure S1). We then exploited the salt-dependency of SYCP3's aggregation state to prepare SYCP3 fibres in solution, which were preserved in vitreous ice.

Electron micrographs showed that SYCP3 fibres in vitreous ice maintained the same macroscopic organization observed by negative-stain EM, consisting of transversal 22 nm-wide striations (figure 1a; electronic supplementary material, figure S2). Different from the ribbon-like appearance of SYCP3 fibres prepared on continuous carbon grids, the SYCP3 fibre embedded in vitreous ice appeared as approximately cylindrical objects, with length in the micrometre scale and a diameter of the oval cross-section of 100–250 nm, indicating that the striation pattern extends three-dimensionally throughout the fibre (figure 1a). To confirm that fibre formation is a robust biochemical property of SYCP3 which persists under different preparation modes, we visualized the SYCP3 fibres by atomic force microscopy (AFM). AFM analysis showed the presence of filamentous SYCP3 fibres that had a clear resemblance to those observed by EM, including the presence of the transverse 22 nm repeat (figure 1b).

As the cryoEM analysis had shown that the SYCP3 fibre maintains its characteristic periodic appearance in solution, we decided to investigate its three-dimensional architecture using cryo-electron tomography. A total of ten tomograms for native SYCP3 fibres were recorded under cryogenic conditions (electronic supplementary material, movie S1) and the tomographic volumes reconstructed (see Material and methods). The presence of a periodic striation pattern was confirmed by Fourier analysis, which revealed a corresponding layer line of spatial frequency of $0.045 \text{ nm}^{-1}$ (22 nm).

## 3.2. Sub-tomogram averaging of the SYCP3 fibre

To determine the structural basis for the homotypic SYCP3 interactions that underpin the three-dimensional architecture of the fibre, we performed sub-tomogram averaging. Due to the apparent dense packing of SYCP3 within the fibre and the low signal-to-noise typical of cryo-electron tomography, we did not attempt to pick individual SYCP3 particles for determination of initial subvolume coordinates. Instead, we chose the clearly identifiable 22 nm striation pattern as the guiding feature for subvolume coordinate selection. Thus, for each striation layer of the fibre, start and end points were selected manually, followed by automatic picking of intervening subvolume coordinate points (electronic supplementary material, figure S3). Two sets of coordinate models were created this way. The first model, which was used to generate an initial reference structure *de novo*, had a large spacing of 44 nm (128 voxels) between each subvolume coordinate along the striation pattern. The second coordinate model featured a narrower spacing of 16.5 nm (48 voxels). This procedure was applied to the ten tomograms, resulting in 5756 subvolume coordinates for set 1 and 46 051 for set 2.

royalsocietypublishing.org/journal/rsob    *Open Biol.* **9**: 190094

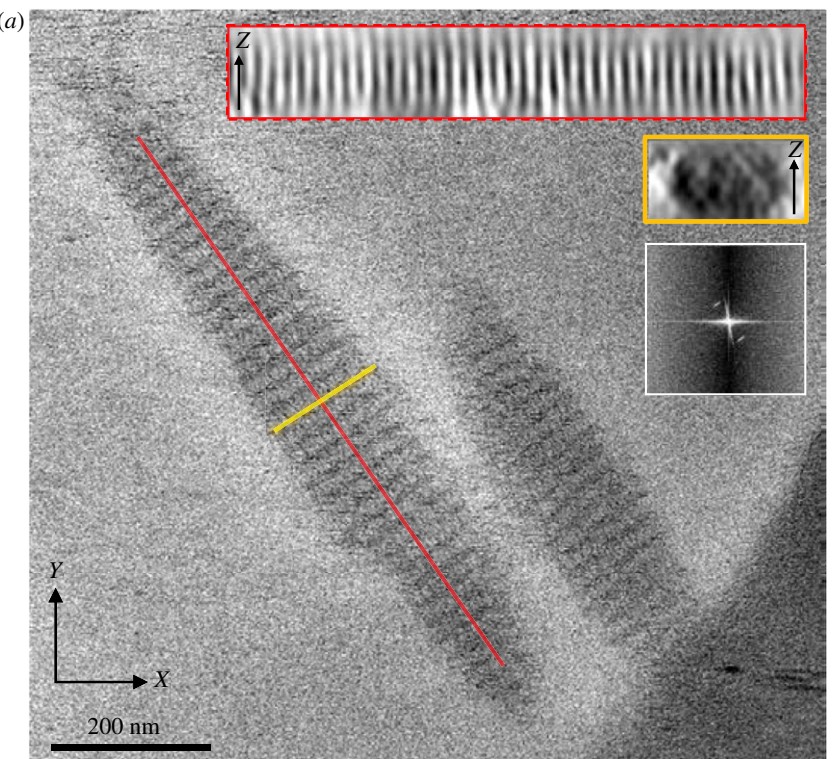

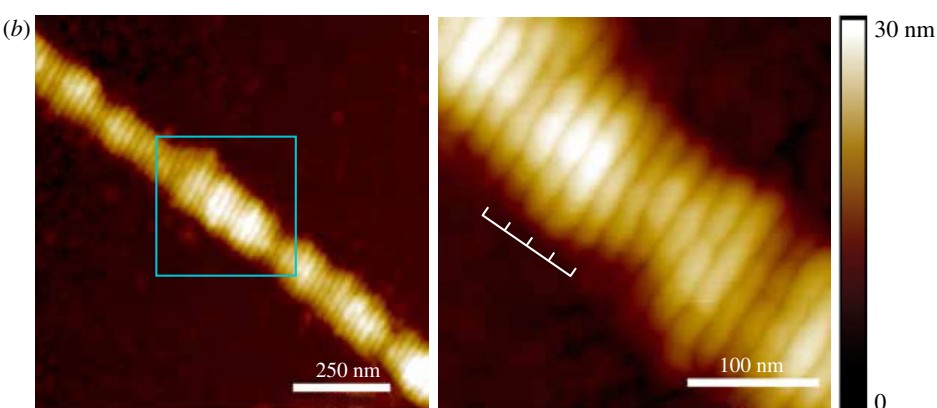

**Figure 1.** Structural analysis of the SYCP3 fibre. (*a*) Cryo-electron micrograph of the SYCP3 fibres, showing the regular pattern of transversal striations in the fibre. The cylindrically shaped fibres measure 0.5–2 μm in length and 100–250 nm in diameter. The insets show two sections of the fibre, along and across its long axis (red and yellow boxes, respectively), and the Fourier analysis of the periodic striation of the fibre, revealing a layer line at a spatial frequency of 0.045 nm$^{-1}$ (22 nm). (*b*) Atomic force microscopy shows that SYCP3 forms filamentous fibres that are similar to those observed by negative-stain electron microscopy [26] and by cryoEM (*a*). The right-hand panel shows a close-up view of the central portion of the fibre (cyan box), highlighting the presence of a transverse repeat of 22 nm.

A reference-free 2D classification of *Z*-projected subvolumes extracted from set 1 was then performed in RELION [41], to obtain 2D averages of the striation pattern and align all particles in the *XY* plane (figure 2*a*). The 2D-class averages indicated that each striation in the fibre contained a parallel array of elongated densities, which resembled the characteristic rod-like shape of the helical core of individual SYCP3 tetramers [26]. The 2D averages further showed that the SYCP3 tetramers are aligned side-by-side in a direction parallel to the fibre long axis, and interact end-to-end to span the 22 nm repeat of the fibre.

To gain further insight into the three-dimensional organization of the SYCP3 fibre, we performed a maximum-likelihood 3D classification using RELION [41]. Exploiting the parallel orientation of the fibres to the grid surface, the subvolumes were aligned based on their projections in the *XY* plane of the fibre. The alignments were then used as the starting point for a

rotational 3D alignment. This procedure resulted in a low-resolution *de novo* structure which served as the initial reference model for masked 3D classification and 3D auto-refinement using the second, larger dataset (electronic supplementary material, figure S4). The final class averages derived from the 3D classification were highly similar, and revealed a remarkable degree of structural heterogeneity in the arrangement of SYCP3 molecules within the fibre (figure 2*b* and electronic supplementary material, figure S5). Importantly, further hierarchical 3D classification did not yield distinct, better-resolved models of the averaged sub-volumes (electronic supplementary material, figure S6) indicating that a lack of long-range 3D order is an intrinsic architectural feature of the SYCP3 fibre.

Sub-tomogram averaging showed that individual rod-shaped SYCP3 tetramers are arranged in a parallel fashion along the long axis of the fibre. Occasionally, a kink or a bend is detected within a particle, in agreement with our previous

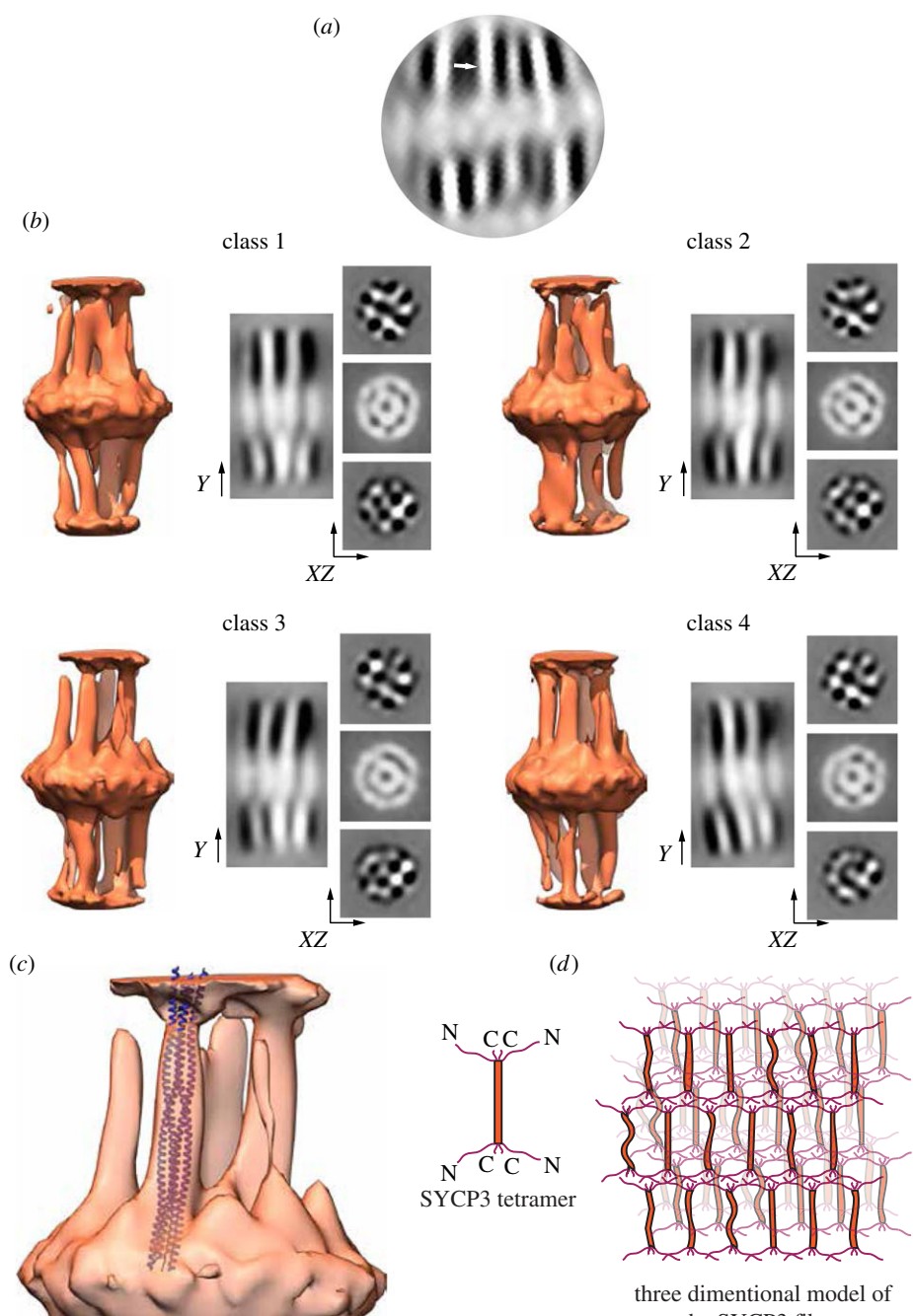

**Figure 2.** Final, refined class averages of the cylindrically masked SYCP3 fibre subtomograms. (*a*) 2D classification of projected subvolumes reveals clear evidence of density connecting adjacent striations (marked by an arrow), consistent with the shape of SYCP3's tetrameric core structure [26]. (*b*) Three-dimensional isosurface rendering of the four refined class averages. The resolution of the final refined classes ranged between 27 and 30 Å as estimated by FSC analysis. The classes had nearly equal distribution and tended to split subvolumes equally regardless of the number of specified classes (electronic supplementary material, figure S5). Isosurface sections along the *Y*-axis and in the *XZ* plane are shown next to each isosurface rendering. The *Y*-axis sections through a three-particle region of the class averages highlight variations in conformation and spacing between individual SYCP3 particles. *XZ* cross-sections through the inter-striation region of all class averages show an approximately rectangular pattern of SYCP3 particles, but with a high degree of heterogeneity in particle arrangement between classes. (*c*) Superposition of the crystal structure of SYCP3 66–230 tetramer (PDB ID 4CPC; shown as blue ribbon) into the isosurface rendering of a refined class average. (*d*) Schematic drawing for the three-dimensional arrangement of SYCP3 molecules in a fibre, based on the result of the cryo-electron tomography. The association between adjacent SYCP3 molecules are mediated exclusively by their intrinsically disordered tails, yielding a highly flexible polymeric assembly.

observation of conformational flexibility in the middle of SYCP3's helical core [26]. Despite lack of side-by-side contacts between adjacent helical cores, the SYCP3 particles remain in vertical register within a layer. Thus, all contacts between SYCP3 particles appear to be mediated by the N- and C-extensions, taking place within the dense striations transversal to the fibre (figure 2*c*). A striking result of the 3D classification is the lack of geometric regularity in the planar arrangement of neighbouring SYCP3 particles, as highlighted by the *XZ* section of

the 3D classes; the sections show only a loosely rectangular arrangement of SYCP3 particles but no discernible long-range order. The implications of this finding for SYCP3 function are analysed in the Discussion.

## 3.3. AFM analysis of the SYCP3-DNA fibres

Alongside its ability to self-associate into a filamentous fibre, SYCP3 possesses a DNA-binding activity, mediated by the

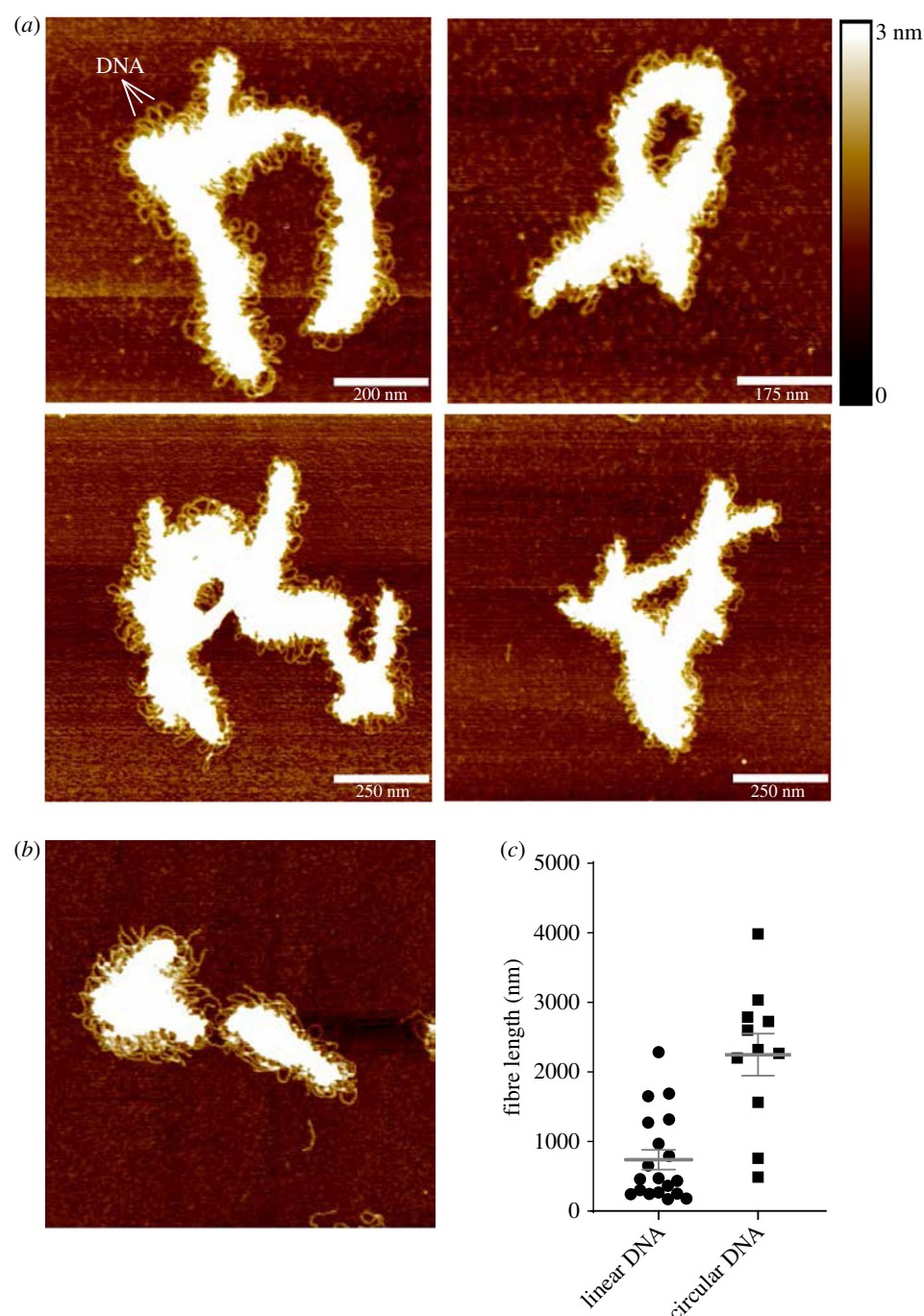

**Figure 3.** Atomic force microscopy of the SYCP3-DNA fibre. (*a*) Representative images of the fibrous structures formed by full-length SYCP3 in the presence of circular plasmid DNA. The DNA is visible as loops protruding from the protein core of the fibre. (*b*) AFM image of SYCP3 bound to linearized plasmid pUC19 DNA. (*c*) Scatter plot of length measurements of SYCP3-DNA fibres that incorporate either linear or circular pUC19 plasmid DNA. Individual values, mean and s.e.m. are shown.

amino acid sequence N-terminal to its helical core. The combination of these two biochemical properties is likely to form the basis for SYCP3's role in lateral element formation. Single-molecule experiments had provided evidence of DNA-dependent clustering by a C-terminally truncated version of SYCP3 that is defective in fibre formation [30]. However, direct evidence that the SYCP3 fibre is competent to interact with DNA has been lacking.

To investigate its potential mode of DNA binding, we prepared samples of SYCP3 for AFM in the presence of plasmid DNA. The AFM images provided striking evidence of DNA incorporation into the SYCP3 fibre (figure 3*a*; electronic supplementary material, figure S7). The regular and continuous presence of DNA loops projecting from the protein

core along the fibre length indicates that an established mechanism of protein–DNA interaction operates within the proteinaceous core of the fibre. The ability of full-length SYCP3 to polymerize in a fibre depends critically on N- and C-terminal sequences flanking its helical core [26], and homotypic interactions were also found to be important for SYCP3-DNA fibre formation: SYCP3 1–230, lacking six residues at its C-terminus that are required for self-association but retaining DNA-binding ability, did not form extended fibres in the presence of DNA. Instead, clusters of DNA-bound SYCP3 1–230 were detected, in agreement with its behaviour in single-molecule optical tweezer experiments (electronic supplementary material, figure S8) [30]. DNA-bound fibres were also observed in the presence of plasmid DNA that had been

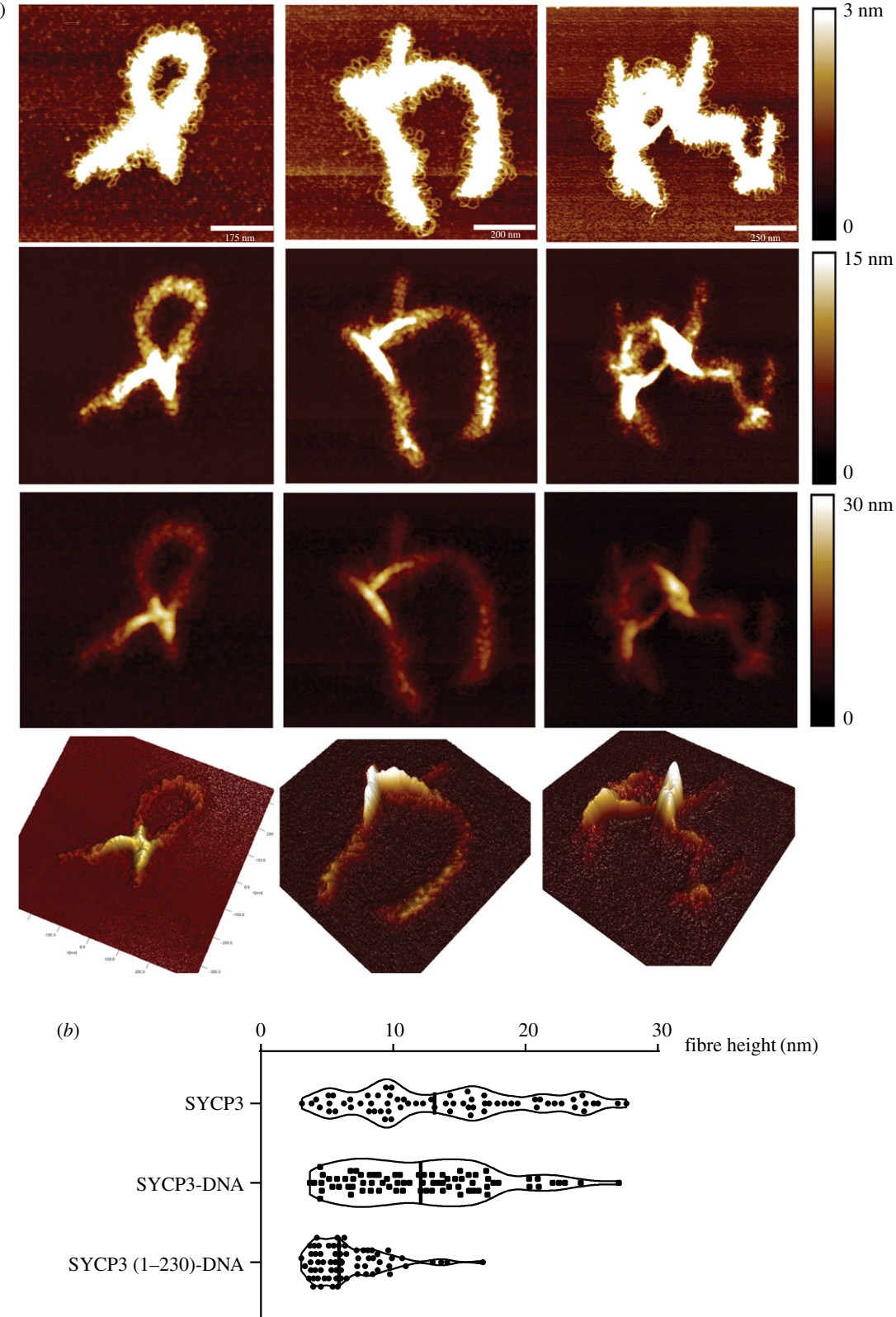

**Figure 4.** Height distribution of the SYCP3 fibres. (*a*) Three examples of DNA-bound SYCP3 fibre, displayed at increasing height threshold from top to bottom to highlight the variable height of the fibre along its axis. The bottom panels show a 3D-representation of each AFM image. (*b*) Violin plots of the measured heights across free SYCP3, SYCP3-DNA and SYCP3 (1–230)-DNA fibres. The DNA is pUC19 plasmid DNA. Individual values and median are shown.

linearized, but were of more limited extent (figure 3*c*; electronic supplementary material, figure S9). The shorter length of the fibre when bound to linear DNA suggests that fibre growth is assisted by a degree of DNA supercoiling, probably because it facilitates DNA bridging by SYCP3.

Inspection of the DNA-bound SYCP3 fibres showed that fibre height was not uniform, with variation along

the fibre length ranging from 5 to 30 nm (figure 4*a*,*b*; electronic supplementary material, figure S10). A similar height distribution was observed for DNA-free SYCP3 (figure 4*b*). Conversely, the height distribution of the DNA-bound SYCP3 1–230 fibre was sharper and clustered at around 10 nm (figure 4*b*). The greater spread and median height of the fibres formed by the full-length protein, in the

royalsocietypublishing.org/journal/rsob    Open Biol. **9**: 190094

presence or absence of DNA, might reflect its ability to form multi-layer assemblies, whereas the height of the SYCP3 (1–230)-DNA layer might reflect the presence of a single, DNA-bound layer, due to the inability of the truncated protein to self-associate.

# 4. Discussion

In this paper, we have provided an experimental description of the SYCP3 fibre, in isolation and bound to DNA. We have shown that the filamentous SYCP3 fibres are based on a loose and heterogeneous mode of packing, and that they can readily incorporate DNA into their structure.

The homotypic interactions of SYCP3 tetramers within the fibre represent a remarkable example of how a large three-dimensional scaffold can be built rapidly and efficiently out of a small number of homotypic interactions between intrinsically disordered protein regions. Thus, the only apparent inter-particle contacts in the fibre are those between N- and C-terminal tails of the SYCP3 tetramers, leaving the coiled-coil core of the SYCP3 tetramer to act as a strut between layers in the fibre. The strut itself appears to be rather flexible, as demonstrated by our EM analysis and the observation of high B factors in the central region of the crystal structure of SYCP3's helical core [26]. The combination of flexible linkages between adjacent SYCP3 molecules and pliability of the SYCP3 helical core endows the fibre with a high degree of plasticity. Thus, rather than a rigid 3D scaffold, the SYCP3 fibre is akin to flexible chain mail.

We note that the peculiar structural properties of the SYCP3 fibre described here seem compatible with the proposed liquid-crystal properties of the SC [42]. The plasticity of the SYCP3 fibre highlighted by our data suggests that, rather than fulfilling a structural role in determining chromosome architecture, SYCP3 might form a protein layer coating the surface of the existing chromosome structure established by meiotic cohesins. The function of such SYCP3 coating might include maintenance of local chromosomal architecture, modulating the availability of defined chromosomal domains during recombination, and acting as a recruiting platform for meiotic factors such as the HORMAD proteins and SYCP2.

The relationship between SYCP3 and SYCP2 in LE assembly appears particularly intimate. The two proteins rely on each other for recruitment to the chromosome axis [16,19,43,44]. Biochemical and structural studies with purified recombinant proteins have shown a potentially significant difference: whereas SYCP3 can exist as both a homo-tetramer [26,45] and a 2 : 2 SYCP3-SYCP2 hetero-tetramer [45], SYCP2 appears to be stable only as a hetero-tetramer with SYCP3 [45]. Taken together, this evidence suggests that SYCP3 might exist in two functionally relevant forms, as a homo-tetramer and a SYCP2-SYCP3 heterotetramer. As both SYCP2 and SYCP3 are coiled-coil proteins, their recruitment to and functional cooperation at the chromosome axis is likely to involve the formation and dynamic rearrangement of homo- and hetero-oligomers. How the spatial and temporal formation of the relevant helical assembly might be regulated is unclear, and it has been suggested that the thermal stability of the coiled-coil assemblies might play a role. The melting temperature of the coiled core region of human SYCP3, spanning amino acids 66–230, that we had previously measured [26] shows

that human SYCP3 homo-tetramers are highly stable assemblies. At the end of meiotic prophase, the SC is rapidly disassembled and SYCP3 disappears from the chromosome axis, except for the centromeric region [46,47]. As formation of the SYCP3 fibre is a fast and irreversible process *in vitro*, SYCP3 removal might require post-translational modifications that weaken its self-association and target it for degradation [48].

Our AFM experiments provide the first direct evidence for the ability of the SYCP3 fibre to form in the presence of DNA. The array of DNA loops protruding from the DNA-bound SYCP3 fibre indicates that, rather than a sporadic phenomenon, the interaction with DNA is a specific property of the SYCP3 fibre. Although our AFM data do not yield dynamic information about the process of fibre formation, our findings suggest that SYCP3 polymerization on the chromosome axis might be coupled to binding chromosomal DNA loops. What is the functional relationship between the DNA-free and DNA-bound states of the SYCP3 fibre? A SYCP3 truncation that abolishes fibre formation limits the size of the DNA-bound fibres (figure 4a), indicating that the DNA-free and DNA-bound states of the fibre share at least some of their homotypic interactions. Conversely, the AFM analysis shows that the DNA-bound fibre lacks the characteristic periodic striations, suggesting that at least some of the underlying interactions between SYCP3 molecules have changed. We did not observe hybrid fibres containing both DNA-bound and DNA-free segments, so their seamless integration might be impossible within the plane of the fibre. However, it is conceivable that DNA-bound and DNA-free forms of the SYCP3 fibre might stack together in a multi-layer structure. The presence of additional SYCP3 on top of a DNA-bound SYCP3 layer might ensure lateral element stability or reinforce the functional insulation of specific chromosomal domains.

Our findings provide a structural and mechanistic basis to help unravel the role of the SC lateral element in meiosis. Future investigations will aim to elucidate the interaction mechanism of the SYCP3 fibre with DNA and with other known components of the meiotic chromosome axis. Such studies will be essential to improve our knowledge of the molecular mechanisms underlying meiotic recombination and the impact on fertility when such mechanisms become defective. The realization that meiotic gene products, including SYCP3, are upregulated in a number of cancers [49,50] further highlights the importance of these studies to understand the pathological mechanisms that contribute to genomic instability in human cells.

Data accessibility. This article has no additional data.

Authors' contributions. D.B., L.R. and L.J. expressed and purified the SYCP3 proteins. D.B. collected the cryo-electron tomography data, assisted by J.M.P., and performed the sub-tomogram averaging analysis of the SYCP3 fibre. L.R., L.J. and I.M. prepared the samples for atomic force microscopy and I.M. performed the imaging and analysis of the data, with advice from R.M.H. D.B., L.R., L.J., I.M. and L.P. designed experiments and analysed results. L.P. conceived the project and directed the research.

Competing interests. We declare we have no competing interests.

Funding. This work was supported by a project grant from the Medical Research Council (grant no. MR/N000161/1) to L.P.

Acknowledgements. We would like to thank Joseph Maman for advice and assistance with the SYCP3 fibre formation assay, and John Heumann and Johanna Syrjanen for their comments and advice.

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
