## [Reviewer comments · Open Biology]

Review History

RSOB-19-0094.R0 (Original submission)

Review form: Reviewer 1

Recommendation

Major revision is needed (please make suggestions in comments)

Are each of the following suitable for general readers?

- a) **Title**
Yes

- b) **Summary**
Yes

c) Introduction

Yes

Is the length of the paper justified?

Yes

Should the paper be seen by a specialist statistical reviewer?

No

Is it clear how to make all supporting data available?

Not Applicable

Is the supplementary material necessary; and if so is it adequate and clear?

Yes

Do you have any ethical concerns with this paper?

No

Comments to the Author

The article authored by Bollschweiler et al. describes the SYCP3 fibre architecture, describes the interaction with the DNA and propose a mechanism for the deposition of SYCP3 onto the DNA. The article is well written and consideration has been taken to make it understandable for an audience of non-specialists. Regardless, although this is for the editor to decide, as a more technical (microscopy) reviewer, I do not find that this topic is of extensive interest for an audience outside the field. Should the article be considered sound for the journal I advise that a major revision is made to better support the conclusions.

Starting from the basics and nowadays standard in cryo-electron microscopy (applicable to subtomogram averaging) I do not see a Fourier Shell correlation plot. This does not only show the resolution but it also provides hints on the alignment procedure.

Along this line, I noticed that the authors provide details about the classification but do not describe if the pool of subtomograms has been previously divided into two subgroups and then independently processed. then FSC from the independent averages should be performed for each class. This would be the bare minimum to accept a class as not the result of noise alignment.

The resolution is somehow limited, possibly due to heterogeneity in the structure, but from experience, it is also possible that the alignment is not perfect. A plot of the subtomogram position after alignment on the tomographic slice colour-coded for inclusion/exclusion from the final average should be shown (not necessarily in the paper but definitely in the response). This check is critical to see what is the behaviour of the alignment considering the high overlap required during the extraction procedure.

Further samples of the row subtomograms representing each class should be shown next to the average to demonstrate how solid the classification is.

The AFM analyses do not really bring anything that a simpler and faster gel would have not provided, further, the comparison of the average fibre length could have been obtained from the cryo-EM data. Personally, I find this distracting and not particularly insightful.

Review form: Reviewer 2

Recommendation

Accept as is

Are each of the following suitable for general readers?

- a) **Title**
Yes
- b) **Summary**
Yes
- c) **Introduction**
Yes

Is the length of the paper justified?

Yes

Should the paper be seen by a specialist statistical reviewer?

No

Is it clear how to make all supporting data available?

No

Is the supplementary material necessary; and if so is it adequate and clear?

Yes

Do you have any ethical concerns with this paper?

No

Comments to the Author

This study describes results of cryo-EM and AFM structural measurements of the SYCP3 protein involved in the lateral element of the synaptonemal complex. SYCP3 has previously been shown to form filamentous structures in the absence of DNA and here the authors confirm that these striated fibres self-assemble. They use tomographic cryo-EM to show that the SYCP3 protein assembles in a parallel fashion via N and C termini interactions to give a 3D structure of the filaments. They use AFM to confirm that these filaments assemble under different in vitro preparation conditions. They go on to use AFM to investigate how SYCP3 assembles with DNA. Overall, they present significant new evidence of the structural role of SYCP3 in the synaptonemal complex.

I am not an expert in cryo-EM so cannot comment on this technique in detail, but the data and interpretation all looks consistent. The AFM data shows that SYCP3 forms fibres on DNA (Figs. 3 and 4) the length of which depends on whether the DNA is linear or circular. Furthermore, the truncated form of SYCP3 missing the C terminus prevents elongated fibres of SYCP3 with DNA forming (Fig. S7), supporting the hypothesis that SYCP3 interacts with itself and DNA through its termini. One of the main conclusions is that the SYCP3 fibres in the presence of DNA no longer shows the regular striated structure. The AFM data shows this outcome very clearly in Figs. 3, 4, S6-9. I am pleased to see in Fig. 4 that they have displayed the AFM at different image thresholds so one can compare the structures adopted by the DNA and the protein. They also include high magnification image of the protein on the DNA in Fig. S6.

Decision letter (RSOB-19-0094.R0)

28-May-2019

Dear Dr Pellegrini,

We are writing to inform you that the Editor has reached a decision on your manuscript RSOB-19-0094 entitled "Molecular architecture of the SYCP3 fibre and its interaction with DNA", submitted to Open Biology.

As you will see from the reviewers' comments below, there are a number of criticisms that prevent us from accepting your manuscript at this stage. The reviewers suggest, however, that a revised version could be acceptable, if you are able to address their concerns. If you think that you can deal satisfactorily with the reviewer's suggestions, we would be pleased to consider a revised manuscript.

The revision will be re-reviewed, where possible, by the original referees. As such, please submit the revised version of your manuscript within four weeks. If you do not think you will be able to meet this date please let us know immediately.

When submitting your revised manuscript, please respond to the comments made by the referee(s) and upload a file "Response to Referees" in "Section 6 - File Upload". You can use this to document any changes you make to the original manuscript. In order to expedite the processing of the revised manuscript, please be as specific as possible in your response to the referee(s).

Please see our detailed instructions for revision requirements
<https://royalsociety.org/journals/authors/author-guidelines/>

Sincerely,
The Open Biology Team
mailto: openbiology@royalsociety.org

Reviewer(s)' Comments to Author(s):

Referee: 1

Comments to the Author(s)

The article authored by Bollschweiler et al. describes the SYCP3 fibre architecture, describes the interaction with the DNA and propose a mechanism for the deposition of SYCP3 onto the DNA.

The article is well written and consideration has been taken to make it understandable for an audience of non-specialists. Regardless, although this is for the editor to decide, as a more technical (microscopy) reviewer, I do not find that this topic is of extensive interest for an audience outside the field. Should the article be considered sound for the journal I advise that a major revision is made to better support the conclusions.

Starting from the basics and nowadays standard in cryo-electron microscopy (applicable to subtomogram averaging) I do not see a Fourier Shell correlation plot. This does not only show the resolution but it also provides hints on the alignment procedure.

Along this line, I noticed that the authors provide details about the classification but do not describe if the pool of subtomograms has been previously divided into two subgroups and then independently processed. then FSC from the independent averages should be performed for each class. This would be the bare minimum to accept a class as not the result of noise alignment.

The resolution is somehow limited, possibly due to heterogeneity in the structure, but from experience, it is also possible that the alignment is not perfect. A plot of the subtomogram position after alignment on the tomographic slice colour-coded for inclusion/exclusion from the final average should be shown (not necessarily in the paper but definitely in the response). This check is critical to see what is the behaviour of the alignment considering the high overlap required during the extraction procedure.

Further samples of the row subtomograms representing each class should be shown next to the average to demonstrate how solid the classification is.

The AFM analyses do not really bring anything that a simpler and faster gel would have not provided, further, the comparison of the average fibre length could have been obtained from the cryo-EM data. Personally, I find this distracting and not particularly insightful.

Referee: 2

Comments to the Author(s)

This study describes results of cyro-EM and AFM structural measurements of the SYCP3 protein involved in the lateral element of the synaptonemal complex. SYCP3 has previously been shown to form filamentous structures in the absence of DNA and here the authors confirm that these striated fibres self-assemble. They use tomographic cryo-EM to show that the SYCP3 protein assembles in a parallel fashion via N and C termini interactions to give a 3D structure of the filaments. They use AFM to confirm that these filaments assemble under different in vitro preparation conditions. They go on to use AFM to investigate how SYCP3 assembles with DNA. Overall, they present significant new evidence of the structural role of SYCP3 in the synaptonemal complex.

I am not an expert in cryo-EM so cannot comment on this technique in detail, but the data and interpretation all looks consistent. The AFM data shows that SYCP3 forms fibres on DNA (Figs. 3 and 4) the length of which depends on whether the DNA is linear or circular. Furthermore, the truncated form of SYCP3 missing the C terminus prevents elongated fibres of SYCP3 with DNA forming (Fig. S7), supporting the hypothesis that SYCP3 interacts with itself and DNA through its termini. One of the main conclusions is that the SYCP3 fibres in the presence of DNA no longer shows the regular striated structure. The AFM data shows this outcome very clearly in Figs. 3, 4, S6-9. I am pleased to see in Fig. 4 that they have displayed the AFM at different image thresholds so one can compare the structures adopted by the DNA and the protein. They also include high magnification image of the protein on the DNA in Fig. S6.

Author's Response to Decision Letter for (RSOB-19-0094.R0)

See Appendix A.

RSOB-19-0094.R1 (Revision)

Review form: Reviewer 1

Recommendation

Major revision is needed (please make suggestions in comments)

Do you have any ethical concerns with this paper?

No

Comments to the Author

The authors have made an effort to include the requested data, but there are a few concerns that have not been removed:

The FSC curves are extremely similar to each other. This is the result of averaging of subtomograms coming from multiple tomograms which I would expect have slightly different defocus values, I would have expected more difference given the class sizes and the obvious difference in the quality of the densities. Were the data CTF corrected? the methods only mention CTF estimation. Which values were used for the frequency filters?

The model files display an extremely rigid structure, the models should be overlaid to a tomogram. Given the differences in the classes, my experience leads me to question the result and to infer that the alignment procedure is not solid. Stating that Relion was used is not satisfactory here, as the result is strictly dependent on the decision made by the user.

The gold standard FSC only reduces the risk to have a reference bias, but the lack of proper alignment cannot be excluded through its use.

Honestly, I do not think those classes are real, most likely they are the result of partial misalignments. I would suggest that if the article is sound only the most represented one is shown, and the text adjusted accordingly.

Decision letter (RSOB-19-0094.R1)

30-Jul-2019

Dear Dr Pellegrini,

We are writing to inform you that the Editor has reached a decision on your manuscript RSOB-19-0094.R1 entitled "Molecular architecture of the SYCP3 fibre and its interaction with DNA", submitted to Open Biology.

As you will see from the reviewer's comments below, there are a number of criticisms that

prevent us from accepting your manuscript at this stage. The reviewer suggests, however, that a revised version could be acceptable, if you are able to address their concerns. If you think that you can deal satisfactorily with the suggestions, we would be pleased to consider a revised manuscript.

The revision will be re-reviewed, where possible, by the original referees. As such, please submit the revised version of your manuscript within four weeks. If you do not think you will be able to meet this date please let us know immediately.

When submitting your revised manuscript, please respond to the comments made by the referee(s) and upload a file "Response to Referees" in "Section 6 - File Upload". You can use this to document any changes you make to the original manuscript. In order to expedite the processing of the revised manuscript, please be as specific as possible in your response to the referee(s).

Please see our detailed instructions for revision requirements. It is essential these instructions are followed carefully to minimize any delay to publication:
<https://royalsociety.org/journals/authors/author-guidelines/>

Sincerely,

The Open Biology Team
mailto: openbiology@royalsociety.org

Reviewer(s)' Comments to Author(s):

Referee:

Comments to the Author(s)

The authors have made an effort to include the requested data, but there are a few concerns that have not been removed:

The FSC curves are extremely similar to each other. This is the result of averaging of subtomograms coming from multiple tomograms which I would expect have slightly different defocus values, I would have expected more difference given the class sizes and the obvious difference in the quality of the densities. Were the data CTF corrected? the methods only mention CTF estimation. Which values were used for the frequency filters?

The model files display an extremely rigid structure, the models should be overlaid to a tomogram. Given the differences in the classes, my experience leads me to question the result and

to infer that the alignment procedure is not solid. Stating that Relion was used is not satisfactory here, as the result is strictly dependent on the decision made by the user.

The gold standard FSC only reduces the risk to have a reference bias, but the lack of proper alignment cannot be excluded through its use.

Honestly, I do not think those classes are real, most likely they are the result of partial misalignments. I would suggest that if the article is sound only the most represented one is shown, and the text adjusted accordingly.

Author's Response to Decision Letter for (RSOB-19-0094.R1)

See Appendix B.

RSOB-19-0094.R2 (Revision)

Review form: Reviewer 1

Recommendation

Major revision is needed (please make suggestions in comments)

Do you have any ethical concerns with this paper?

No

Comments to the Author

I continue to have reservations about the quality of the alignment but most specifically about the validity of the classification. I would remove that section. I have had this kind of results in the past and they are generally due to misalignments. If this is not the case and the authors are convinced of the correctness of the classification the paper should have a figure showing an exemplary subtomogram per class next to each class average (i would add it to S6), the data should be available therefore this should be easy to demonstrate.

Decision letter (RSOB-19-0094.R2)

27-Aug-2019

Dear Dr Pellegrini

We are pleased to inform you that your manuscript RSOB-19-0094.R2 entitled "Molecular architecture of the SYCP3 fibre and its interaction with DNA" has been accepted by the Editor for publication in Open Biology. The reviewer(s) have recommended publication, but also suggest

some minor revisions to your manuscript. Therefore, we invite you to respond to the reviewer(s)' comments and revise your manuscript.

Please submit the revised version of your manuscript within 14 days. If you do not think you will be able to meet this date please let us know immediately and we can extend this deadline for you.

- 1) A text file of the manuscript (doc, txt, rtf or tex), including the references, tables (including captions) and figure captions. Please remove any tracked changes from the text before submission. PDF files are not an accepted format for the "Main Document".
- 2) A separate electronic file of each figure (tiff, EPS or print-quality PDF preferred). The format should be produced directly from original creation package, or original software format. Please note that PowerPoint files are not accepted.
- 3) Electronic supplementary material: this should be contained in a separate file from the main text and meet our ESM criteria (see <http://royalsocietypublishing.org/instructions-authors#question5>). All supplementary materials accompanying an accepted article will be treated as in their final form. They will be published alongside the paper on the journal website and posted on the online figshare repository. Files on figshare will be made available approximately one week before the accompanying article so that the supplementary material can be attributed a unique DOI.

Online supplementary material will also carry the title and description provided during submission, so please ensure these are accurate and informative. Note that the Royal Society will not edit or typeset supplementary material and it will be hosted as provided. Please ensure that the supplementary material includes the paper details (authors, title, journal name, article DOI). Your article DOI will be 10.1098/rsob.2016[last 4 digits of e.g. 10.1098/rsob.20160049].

- 4) A media summary: a short non-technical summary (up to 100 words) of the key findings/importance of your manuscript. Please try to write in simple English, avoid jargon, explain the importance of the topic, outline the main implications and describe why this topic is newsworthy.

Images

Data-Sharing

It is a condition of publication that data supporting your paper are made available. Data should be made available either in the electronic supplementary material or through an appropriate repository. Details of how to access data should be included in your paper. Please see <http://royalsocietypublishing.org/site/authors/policy.xhtml#question6> for more details.

Data accessibility section

Sincerely,

The Open Biology Team

<mailto:openbiology@royalsociety.org>

Editor's comment:

Please respond to the precise request of the reviewer.

Thanks

Reviewer(s)' Comments to Author:

Referee: 1

Comments to the Author(s)

I continue to have reservations about the quality of the alignment but most specifically about the validity of the classification. I would remove that section. I have had this kind of results in the past and they are generally due to misalignments. If this is not the case and the authors are convinced of the correctness of the classification the paper should have a figure showing an exemplary subtomogram per class next to each class average (i would add it to S6), the data should be available therefore this should be easy to demonstrate.

Decision letter (RSOB-19-0094.R3)

20-Sep-2019

Dear Dr Pellegrini

We are pleased to inform you that your manuscript entitled "Molecular architecture of the SYCP3 fibre and its interaction with DNA" has been accepted by the Editor for publication in Open Biology.

Article processing charge

Please note that the article processing charge is immediately payable. A separate email will be sent out shortly to confirm the charge due. The preferred payment method is by credit card; however, other payment options are available.

Sincerely,

The Open Biology Team
mailto: openbiology@royalsociety.org

Appendix A

Reviewer(s)' Comments to Author(s):

Referee: 1

Starting from the basics and nowadays standard in cryo-electron microscopy (applicable to subtomogram averaging) I do not see a Fourier Shell correlation plot. This does not only show the resolution but it also provides hints on the alignment procedure.

The Fourier shell curves for the final 4 3D classes in Figure 2 are now provided in Supplementary figure 5.

Along this line, I noticed that the authors provide details about the classification but do not describe if the pool of subtomograms has been previously divided into two subgroups and then independently processed. then FSC from the independent averages should be performed for each class. This would be the bare minimum to accept a class as not the result of noise alignment.

The 3D autorefinement carried out in Relion implements a 'gold standard' protocol that is specifically designed to minimize overfitting of the data*.

*2016 Processing of Structurally Heterogeneous Cryo-EM Data in RELION. **579**, 125–157. (doi:10.1016/bs.mie.2016.04.012)

The resolution is somehow limited, possibly due to heterogeneity in the structure, but from experience, it is also possible that the alignment is not perfect. A plot of the subtomogram position after alignment on the tomographic slice colour-coded for inclusion/exclusion from the final average should be shown (not necessarily in the paper but definitely in the response). This check is critical to see what is the behaviour of the alignment considering the high overlap required during the extraction procedure.

We have attached 10 model files containing the subtomogram coordinates for each of the four classes per individual tomogram volume. The model files were generated in IMOD (<https://bio3d.colorado.edu/imod/>). Due to the maximum-likelihood approach implemented in Relion, no discrete inclusion or exclusion positions will be visible for the vast majority of coordinates. It should be emphasised that excessive shifts for aligned subtomograms were removed as described, to prevent the chance of autocorrelation for overlapping subvolumes.

Further samples of the row subtomograms representing each class should be shown next to the average to demonstrate how solid the classification is.

This information was provided in the original manuscript. Supplementary figure 6 (Suppl fig 5 in the original manuscript) shows further classification of each of the 4 four classes into 8 subclasses, highlighting how further classification didn't yield structurally different classes.

The AFM analyses do not really bring anything that a simpler and faster gel would have not provided, further, the comparison of the average fibre length could have been obtained from the cryo-EM data. Personally, I find this distracting and not particularly insightful.

We respectfully disagree with the reviewer. The large size of the SYCP3-DNA fibres makes gel electrophoresis unsuitable as a means of analysis. Indeed, the

visualization of the SYCP3-DNA fibres by AFM, as proof of their biochemical existence, is a considerable experimental achievement.

The SYCP3 fibre samples analysed by cryoEM do not contain DNA, so they could not have been used to measure the average length of the SYCP3-DNA fibres.

Referee: 2

Comments to the Author(s)

This study describes results of cryo-EM and AFM structural measurements of the SYCP3 protein involved in the lateral element of the synaptonemal complex. SYCP3 has previously been shown to form filamentous structures in the absence of DNA and here the authors confirm that these striated fibres self-assemble. They use tomographic cryo-EM to show that the SYCP3 protein assembles in a parallel fashion via N and C termini interactions to give a 3D structure of the filaments. They use AFM to confirm that these filaments assemble under different in vitro preparation conditions. They go on to use AFM to investigate how SYCP3 assembles with DNA. Overall, they present significant new evidence of the structural role of SYCP3 in the synaptonemal complex.

I am not an expert in cryo-EM so cannot comment on this technique in detail, but the data and interpretation all looks consistent. The AFM data shows that SYCP3 forms fibres on DNA (Figs. 3 and 4) the length of which depends on whether the DNA is linear or circular. Furthermore, the truncated form of SYCP3 missing the C terminus prevents elongated fibres of SYCP3 with DNA forming (Fig. S7), supporting the hypothesis that SYCP3 interacts with itself and DNA through its termini. One of the main conclusions is that the SYCP3 fibres in the presence of DNA no longer shows the regular striated structure. The AFM data shows this outcome very clearly in Figs. 3, 4, S6-9. I am pleased to see in Fig. 4 that they have displayed the AFM at different image thresholds so one can compare the structures adopted by the DNA and the protein. They also include high magnification image of the protein on the DNA in Fig. S6.

Nothing to address.

Appendix B

Comments to the Author(s)

The FSC curves are extremely similar to each other. This is the result of averaging of subtomograms coming from multiple tomograms which I would expect have slightly different defocus values, I would have expected more difference given the class sizes and the obvious difference in the quality of the densities.

Gold standard FSC ensures that correlation beyond a given low pass filter (in this case 30 Å) is free from initial model bias. While this of course cannot ensure correct alignment, the fact that consistent structures are determined independently provides strong evidence in support of our model of the SYCP3 fibre. The shape of the FSC curve is a measure of the level of correlation at each Fourier shell; the resolution of the structures is approximately at the first node of the CTF, so similar shapes can be expected. Dips in the FSC caused by defocus-dependent effects are generally low in subtomogram averaging, as the tilted images provide a large defocus range given a single field of view.

The expectation maximization approach used in RELION for averaging allows for each subtomogram to maintain multiple class occupancies and orientations, weighted by the determined Bayesian probabilities. In our case, since each structure has repeating sub-components that are present in different arrangements in each class, it would be expected that each subtomogram appears in multiple classes.

Were the data CTF corrected? the methods only mention CTF estimation.

Yes, the CTF correction was performed in Relion with CTFFIND4.0.15, following the procedure described in Bharat and Scheres, *Nature Protocols*, 2016 (ref. 38). We have now clarified the wording in the relevant section of the Methods.

Which values were used for the frequency filters?

Relion determines an optimised 3D linear filter based on a Bayesian statistical framework. We don't know how to extract (and/or interpret) the numerical value of the filter.

The model files display an extremely rigid structure, the models should be overlayed to a tomogram. Given the differences in the classes, my experience leads me to question the result and to infer that the alignment procedure is not solid. Stating that Relion was used is not satisfactory here, as the result is strictly dependent on the decision made by the user.

The model files attached with the revised manuscript showed the original XYZ positions of the initial subvolume extraction, hence the impression of rigidity. We have now attached the model files (CoordinateShifts.zip folder) with the combined classes for each individual tomogram showing the final shifts after alignment in X and Y (we chose not to perform Z alignment to avoid possible sub-volume overlapping).

The shifts demonstrate that our alignment method is working, as each class has slightly different origins around each of the initial extraction point. Although the shifts are likely to be of limited accuracy due to the lack of high resolution information, we can reliably say that there is no evidence for systematic error in the alignment procedure.

Honestly, I do not think those classes are real, most likely they are the result of partial misalignments. I would suggest that if the article is sound only the most represented one is shown, and the text adjusted accordingly.

For the point about possible misalignment, please see the previous reply. We believe that it is important to show the 4 3D classes of Figure 2, and not simply one class, as they serve the purpose of demonstrating that we found no evidence in the data for a class of SYCP3 fibres with a higher degree of order or differently organised, and that all classes are similarly heterogeneous, as also proven by the further classification in Supplementary figure 6.